# Puromycin reactivity does not accurately localize translation at the subcellular level

Syed Usman Enam[1,2†], Boris Zinshteyn[1,2†], Daniel H Goldman[1,2†], Madeline Cassani[1,2], Nathan M Livingston[3], Geraldine Seydoux[1,2], Rachel Green[1,2]*

[1]Department of Molecular Biology and Genetics, Johns Hopkins University School of Medicine, Baltimore, United States; [2]Howard Hughes Medical Institute, Baltimore, United States; [3]Department of Biophysics and Biophysical Chemistry, Johns Hopkins University School of Medicine, Baltimore, United States

**Abstract** Puromycin is a tyrosyl-tRNA mimic that blocks translation by labeling and releasing elongating polypeptide chains from translating ribosomes. Puromycin has been used in molecular biology research for decades as a translation inhibitor. The development of puromycin antibodies and derivatized puromycin analogs has enabled the quantification of active translation in bulk and single-cell assays. More recently, in vivo puromycylation assays have become popular tools for localizing translating ribosomes in cells. These assays often use elongation inhibitors to purportedly inhibit the release of puromycin-labeled nascent peptides from ribosomes. Using in vitro and in vivo experiments in various eukaryotic systems, we demonstrate that, even in the presence of elongation inhibitors, puromycylated peptides are released and diffuse away from ribosomes. Puromycylation assays reveal subcellular sites, such as nuclei, where puromycylated peptides accumulate post-release and which do not necessarily coincide with sites of active translation. Our findings urge caution when interpreting puromycylation assays in vivo.

*For correspondence:
ragreen@jhmi.edu

†These authors contributed equally to this work

## Introduction

Puromycin is a potent translational inhibitor that binds to ribosomes from all domains of life and has been used as a chemical probe and selectable marker for decades (*Aviner, 2020*; *Yarmolinsky and Haba, 1959*). Puromycin is unique among translational inhibitors in that it is itself a substrate of the ribosomal peptidyl-transferase reaction (*Nathans, 1964*). Puromycin mimics the 3′ adenosine of a tRNA charged with a modified tyrosine, which binds in the ribosomal acceptor site (*Figure 1A*). The ribosome transfers the nascent peptide chain on the P-site tRNA to puromycin, leading to spontaneous dissociation of the nascent peptide from the ribosome (*Figure 1B*; *Nathans, 1964*).

The development of anti-puromycin antibodies and of derivatized analogs of puromycin (*Fujiwara et al., 1982*; *Liu et al., 2012*) has led to the commonplace use of puromycin as a metabolic probe to measure the extent of active translation, replacing radioactive tracers such as S35 methionine. These probes can be used to quantify the amount of active translation from cells in a culture dish, tissue or organism (*Schmidt et al., 2009*; *Chao et al., 2012*). Subsequent development of the ribopuromycylation method (RPM) (*David et al., 2012*; *Bastide et al., 2018*) pushed the technique a step further, claiming to detect the subcellular localization of actively translating ribosomes using a puromycin-specific antibody. In these initial publications, the authors argued that the translation elongation inhibitors cycloheximide or emetine prevent dissociation of the puromycylated peptides from the ribosome. Cycloheximide and emetine, however, bind in the E-site of the ribosome (*Figure 1C*) far from the peptidyl-transferase center (*Garreau de Loubresse et al., 2014*; *Wong et al., 2014*). Previous work (*Grollman, 1968*; *Colombo et al., 1965*) established that these inhibitors prevent puromycin-induced splitting of ribosomes into individual subunits, but do not

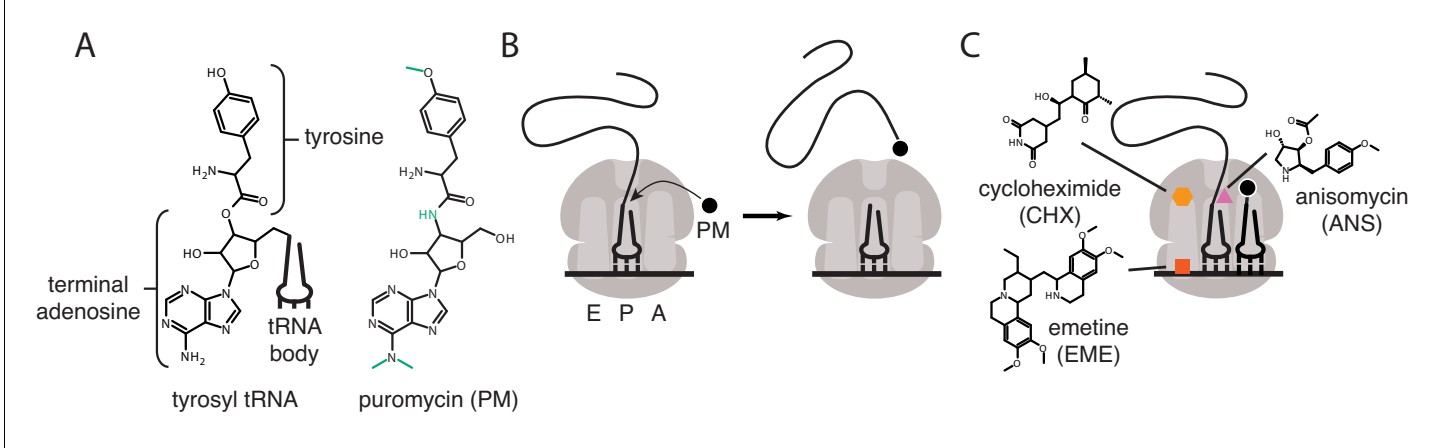

**Figure 1.** Mechanisms of action of puromycin and other translational inhibitors. (**A**) Comparison of structure of 3′ terminus of tyrosyl tRNA with that of puromycin. Key differences are highlighted in green. tRNA body not drawn to scale. (**B**) Scheme for reaction of puromycin with peptidyl P-site tRNA on the ribosome, leading to dissociation of puromycylated peptide. (**C**) Structures and schematicized ribosome binding sites of translational inhibitors cycloheximide, anisomycin and emetine. Binding sites are based on *Garreau de Loubresse et al., 2014*; *Wong et al., 2014*.

prevent the release of the majority of puromycylated peptides. Cycloheximide and emetine, in fact, are sometimes omitted from in vivo puromycylation assays based on the claim that short (approximately 5 min) labeling times capture peptides near their original site of translation before significant diffusion has taken place (*tom Dieck et al., 2015*; *Biever et al., 2020*; *Lewis et al., 2018*; *Glock et al., 2020*; *Holt et al., 2019*). Puromycin-based imaging methods have been widely adopted, particularly in neurobiology, where translation in neuronal processes, far from the cell body, is crucial to neuronal function (*tom Dieck et al., 2015*; *Biever et al., 2020*; *Lewis et al., 2018*; *Glock et al., 2020*; *Holt et al., 2019*; *V'kovski et al., 2019*; *Langille et al., 2019*; *Gonatopoulos-Pournatzis et al., 2020*; *Graber et al., 2013*; *Graber et al., 2017*). Some studies have combined puromycin treatment with proximity-dependent ligation (PLA) to monitor the location of translation of a specific protein (*tom Dieck et al., 2015*), but this method again does not address diffusion of puromycylated peptides post-release from the ribosome.

In the present work, we establish that puromycin-based methods, as currently implemented, do not accurately localize translation at the subcellular level. We used a rabbit reticulocyte lysate system to show that puromycin nearly instantaneously releases nascent proteins from the ribosome, and that this release reaction is completely unaffected by emetine. To validate this finding in cells, we visualized sites of active translation using fixed cell single-molecule imaging with the SunTag reporter system. Brief treatment with puromycin nearly completely dissociated nascent peptides from their mRNAs, again, independent of the presence of emetine. Simple diffusion calculations predict that the released peptides could diffuse to nearly any point within even large mammalian cells within seconds or minutes. Thus, puromycylation methods described in the literature do not establish subcellular localization of translation.

## Results

### The puromycin analog OPP labels nuclei in live *C. elegans* germlines in the presence or absence of emetine

O-propargyl-puromycin (OPP) is a click-reactive cell permeable puromycin analog that is commonly used to localize sites of translation (*Liu et al., 2012*). When incubated with live cells or tissues, OPP reacts with translating ribosomes and becomes covalently attached to elongating peptides. Post-labeling, OPP is detected by click-reactive chemistry which attaches a fluorescent probe to OPP (*Figure 2*). Using this method to label translation in live *C. elegans* gonads, we observed bright labeling of live germlines upon a 5 min incubation with OPP. The OPP signal was most intense in nuclei, specifically in the chromatin-free center where nucleoli reside. A lower signal was also observed in the

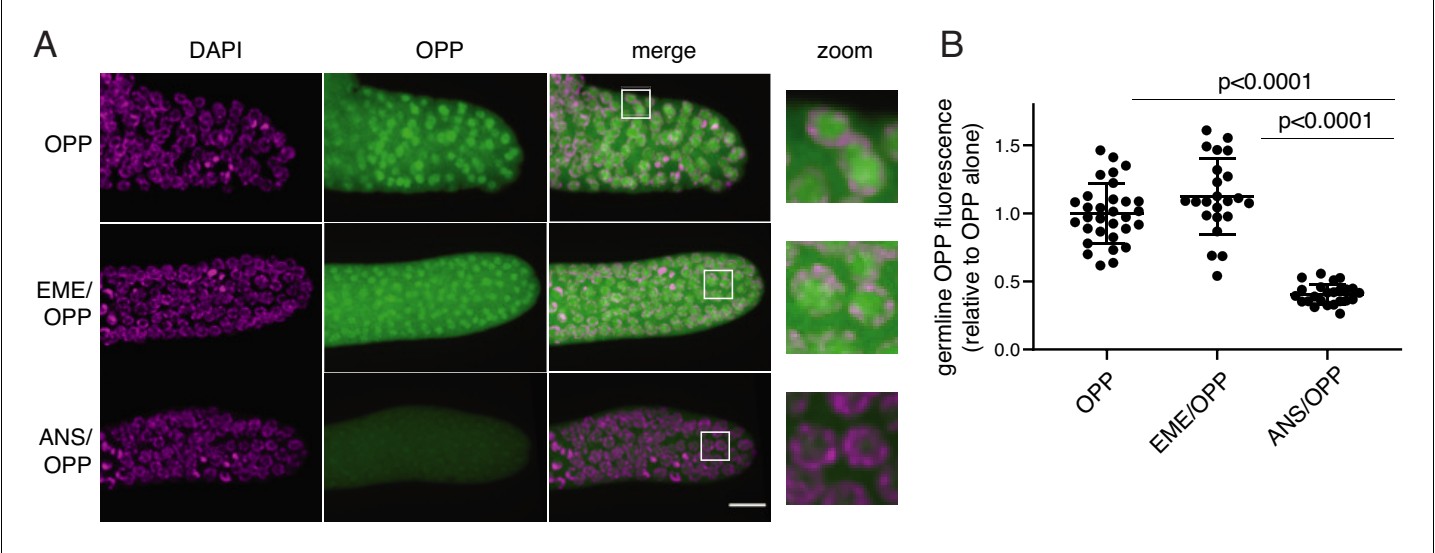

**Figure 2.** O-propargyl-puromycin (OPP) labels nuclei in the distal germline of *C. elegans* in the presence or absence of emetine. (**A**) Representative photomicrographs of germlines labeled for 5 min with 20 µM OPP, and pre-treated for 15 min with control buffer (top row), 45 µM emetine (second row), or 37 µM anisomycin (bottom row). DAPI labels chromosomes. Post-fixation, click labeling of OPP with Alexa Fluor 488 picolyl azide revealed OPP throughout the cytoplasm and concentrated in nuclei. Scale bar = 10 µm. (**B**) Quantification of OPP-Alexa 488 signal in distal germlines. Each dot represents the average fluorescence of the mitotic zone of one worm germline. Values are normalized to the average obtained for germlines pre-treated with control buffer (OPP alone). P values were obtained through an unpaired t-test. Experiment performed in duplicate.

The online version of this article includes the following source data for figure 2:

**Source data 1.** Source data for *Figure 2B*.

cytoplasm, which contains the majority of (if not all) functional ribosomes (*Klinge and Woolford, 2019*). OPP labeling of nuclei was ablated by pre-treatment with anisomycin, a competitive inhibitor of puromycin that stops elongation by binding to the peptidyl-transferase center (*Grollman, 1967*), thereby preventing puromycin from reacting with the nascent chain. In contrast, OPP labeling was unaffected by pre-treatment with emetine (*Figure 2B*). Emetine-resistant puromycin labeling of nucleoli has been observed previously in tissue culture cells (*David et al., 2012*) and may reflect trafficking or diffusion of puromycylated peptides into the nucleolus (*Kubota et al., 1999*; *Schmidt et al., 1995*). We conclude that OPP labels translational products but does not necessarily identify sites of active translation even in the presence of emetine.

## Emetine does not prevent release of puromycylated peptides in rabbit reticulocyte lysates

To determine whether emetine prevents release of puromycylated nascent peptides in vitro, we made use of a previously established real-time translation monitoring assay in rabbit reticulocyte lysate (RRL). This method relies on the fact that luciferase rapidly folds into an enzymatically active conformation only after release from the ribosome (*Kolb et al., 1994*; *Frydman et al., 1994*). By programming RRL with a luciferase mRNA that is truncated (by runoff SP6 transcription of restriction-digested plasmid) just upstream of the stop codon, we accumulate stalled ribosomes at the 3' end of the mRNA, in which the luciferase nascent peptide remains ribosome-bound and enzymatically inactive (*Figure 3A*). RRL programmed with the truncated mRNA (yellow trace) displays little luciferase activity compared to RRL translating full-length mRNA (purple trace). Addition of 91 µM puromycin (the same concentration used by *David et al., 2012*) causes a sharp increase in luminescence output from the truncated mRNA, consistent with release of the stalled peptides.

We reasoned that if emetine prevents release of puromycylated nascent peptides, then emetine should block the increase in luminescence observed upon puromycin addition. Matching the conditions of *David et al., 2012*, we treated the RRL with 208 µM emetine and 91 µM puromycin. When added separately, these two drugs effectively inhibit translation of full-length luciferase mRNA encoding a normal stop codon (*Figure 3—figure supplement 1A*; *David et al., 2012*). Upon

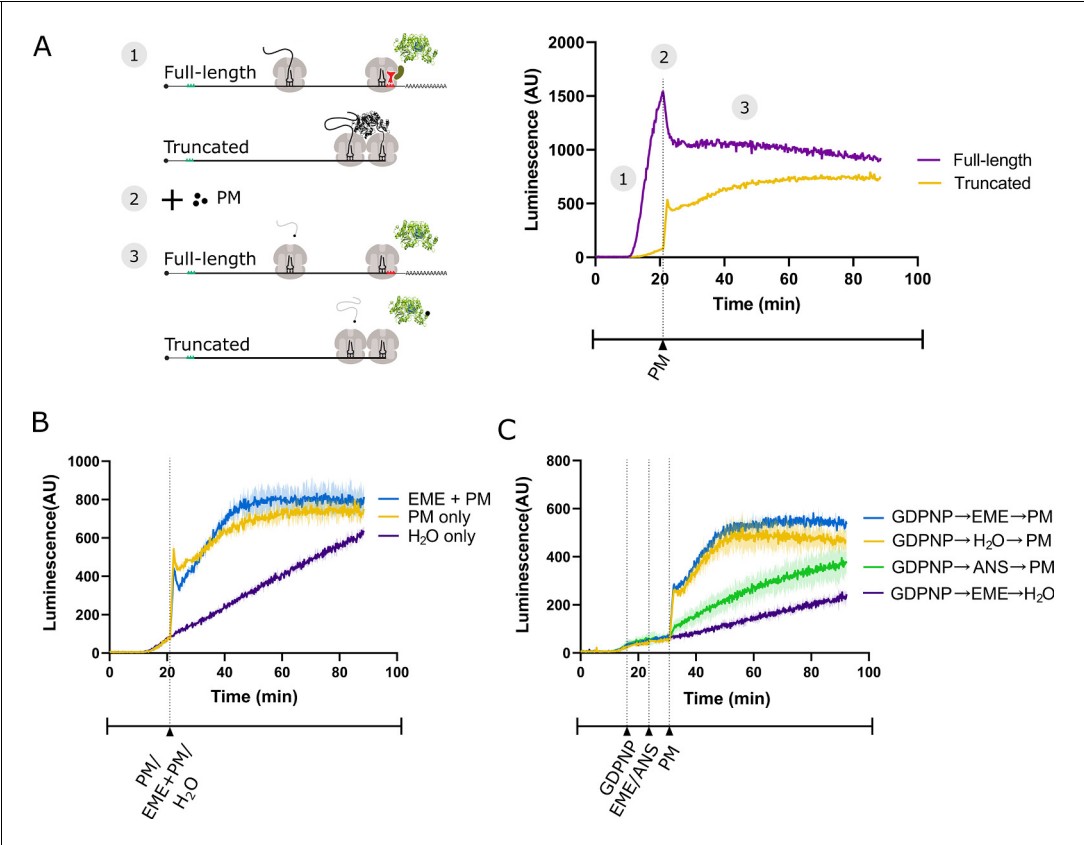

**Figure 3.** Emetine does not prevent release of puromycylated luciferase from rabbit reticulocyte ribosomes. (**A**) Schematic of the real-time translation monitoring assay in rabbit reticulocyte lysate. (1) (Purple trace) Ribosomes translate the full-length luciferase mRNA and release luciferase which becomes enzymatically active and results in an increase in luminescence. (Yellow trace) Ribosomes stall at the 3' end of a truncated luciferase mRNA and little to no luminescence is observed as the ribosome-bound luciferase peptides are in an enzymatically inactive conformation. (2) Puromycin (PM) is added to the system, stopping further translation and causing all nascent peptides to release from the ribosomes. (3) (Yellow) The luciferase rapidly folds into an enzymatically active conformation and a substantial increase in luminescence is observed. (**B**) Either puromycin (yellow), $H_2O$ (purple) or a mixture of emetine (EME) and puromycin (blue) was added to a reaction containing truncated luciferase mRNA at $t = 21$ min. Experiment was performed in duplicate; mean traces shown as solid lines and range of replicates shaded. (**C**) GDPNP was added to a reaction containing truncated luciferase mRNA at $t = 16$ min for 5 min to inhibit translation across samples. Then, either emetine (blue, purple), anisomycin (ANS) (green) or $H_2O$ (yellow) was added to the reaction followed by puromycin (blue, yellow, green) or $H_2O$ (purple) 5 min later. Experiment was performed in duplicate; mean traces shown as solid lines and range of replicates shaded. Note that the experiments in (**A and B**), and *Figure 3—figure supplement 1B* were done in the same batch, and the yellow traces (PM treated) in these panels are the same.

The online version of this article includes the following source data and figure supplement(s) for figure 3:

**Source data 1.** Source data for *Figure 3A, B and C*.

**Figure supplement 1.** Additional control experiments for lysate-based luciferase assays.

**Figure supplement 1—source data 1.** Source data for *Figure 3—figure supplement 1A and B*.

addition of puromycin to lysate programmed with truncated mRNA, we noticed the expected steep increase in luminescence (yellow trace) that was not inhibited by simultaneous addition of emetine (blue trace) (*Figure 3B*). The luminescence of the no-puromycin control (purple trace) increased slowly over time, likely due to low levels of ribosome rescue activity (*Shao et al., 2013*) or spontaneous peptidyl-tRNA hydrolysis in the lysate.

We next considered the possibility that blocking peptide release with emetine requires pre-incubation. To test this, we pre-treated the lysate with emetine 5 min before addition of puromycin. Because pre-treatment would decrease the total translation time and overall luminescence of a sample, it was critical to equalize the total uninhibited reaction time of all samples. This was accomplished in two different ways. In a first experiment, we treated all samples with the nonhydrolyzable GTP analog 5'-guanylyl imidodiphosphate (GDPNP) to inhibit the translational GTPases and prevent

ongoing translation while leaving the ribosome free to react with emetine and puromycin (*Figure 3C*). Again, puromycin treatment (yellow trace) caused a sharp increase in luminescence that was not affected by emetine pre-treatment (blue trace) but was inhibited by anisomycin pre-treatment (green trace). The residual slow increase in the anisomycin trace is likely due to incomplete inhibition by anisomycin resulting from its stochastic dissociation during the reaction. In a second experiment, we added puromycin for the puromycin-only control at the same time that we started pretreating the other samples with inhibitors. This effectively inhibited translation in all samples at the same time (*Figure 3—figure supplement 1B*). While the increase in luminescence for the puromycin-only control (yellow trace) occurred earlier than for the pretreated samples, once the puromycin was added, the luminescence activity of the emetine pretreated sample (blue trace) matched that of the puromycin-only control. Taken together, these results show that pretreating translating ribosomes with emetine does not prevent the release of nascent peptides by puromycin in vitro.

## Emetine does not prevent release of puromycylated peptides in cells

To directly test whether emetine blocks release of puromycylated nascent chains in vivo, we implemented the SunTag method for monitoring translation on single mRNAs (*Pichon et al., 2016*; *Wu et al., 2016*; *Morisaki et al., 2016*; *Wang et al., 2016*; *Yan et al., 2016*). This technique relies on a reporter mRNA encoding tandem repeats of the SunTag epitope near the 5′ end of the coding sequence (*Figure 4A*). When translated, each SunTag peptide is bound by a single chain variable fragment (scFV) of a GCN4 antibody fused to super folder GFP (scFV-sfGFP). An auxin-inducible degron (AID) near the 3′ end of the coding sequence allows controlled degradation of the fully-synthesized SunTag array, reducing fluorescence background and enabling detection of single fully-synthesized polypeptides. We performed fixed-cell imaging of U-2OS cells stably expressing both the SunTag reporter and scFV-sfGFP, detecting mRNA by fluorescence in-situ hybridization (FISH) and SunTag signal by immunofluorescence (IF). With this single-molecule FISH and IF hybrid assay (smFISH-IF), we quantified the association of SunTag nascent chains with their encoding mRNAs under various treatment conditions.

In untreated cells, an average of 63% of single mRNAs (red foci) per cell co-localize with bright SunTag signal (green foci) (*Figure 4B*, top row and 4C); these co-localized spots reflect mRNAs bound by ribosomes synthesizing the SunTag reporter, while weaker isolated green spots reflect single fully synthesized SunTag polypeptides that have been released from the ribosome (*Wu et al., 2016*). Upon treatment with 91 µM puromycin for 5 min, an average of only 3% of mRNAs per cell colocalize with green signal, consistent with release of nascent chains upon puromycin treatment. Remarkably, pre-treatment with 208 µM emetine for 15 min yielded similar results: only 5% of mRNAs on average colocalized with SunTag signal. Importantly, pre-treatment for 5 min with elongation inhibitor anisomycin (37 µM), resulted in an average of 50% of mRNAs co-localized with green foci, as seen in untreated cells. Together, these data indicate that 5 min of puromycin treatment causes release of nascent polypeptides and diffusion away from ribosomes. Pre-treatment with emetine has no effect on puromycin-induced release.

## Puromycylation treatment times are long compared to protein diffusion rates

While initial reports argued that emetine was required to stabilize the interaction of puromycylated peptides with ribosomes, some recent studies of local protein synthesis via the puromycylation method relied on treatment with puromycin alone for ~5–10 min, with the implication that detected nascent proteins do not appreciably diffuse away from their site of synthesis (i.e. ribosome) within the treatment time (*Colombo et al., 1965*; *tom Dieck et al., 2015*; *Morisaki et al., 2016*). To determine how far a nascent protein might diffuse on these timescales (i.e. the spatial resolution of the method), we calculated the expected displacement as a function of time, based on the previously measured diffusion coefficient of GFP in the cytosol (*Di Rienzo et al., 2014*; *Figure 5*). This calculation depends on the dimensionality of space in which the molecule is confined. However, even in the most limiting case of one-dimensional diffusion—approximating movement along a very narrow neural projection—a protein is expected to diffuse ~100 µm in less than 1 min. This distance is large compared to both the scale of the relevant structures to which protein synthesis was localized in neurons (tens of microns) (*tom Dieck et al., 2015*; *Biever et al., 2020*), and to the diameter of HeLa

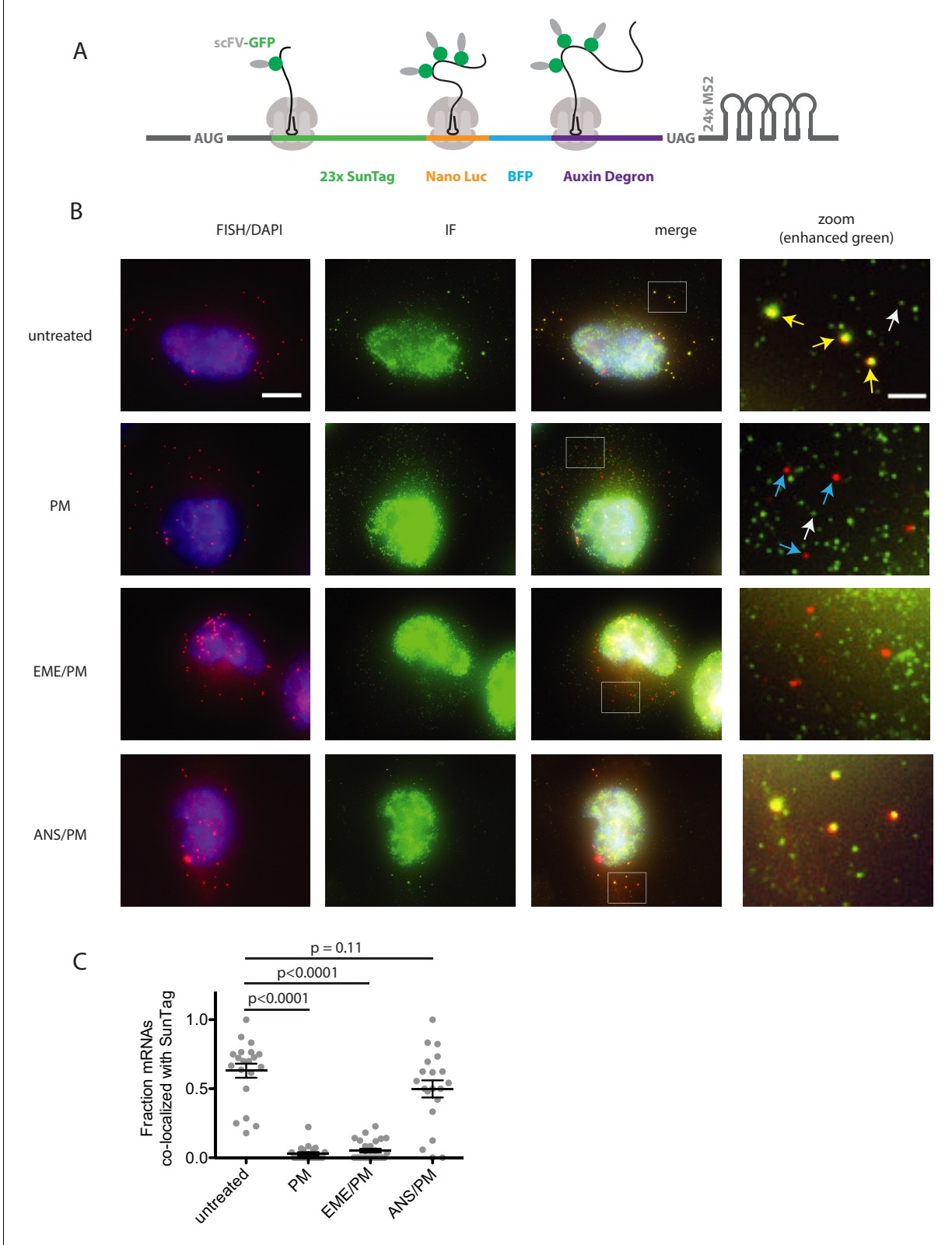

**Figure 4.** Puromycin treatment causes loss of nascent peptide-mRNA co-localization, independent of elongation inhibitors. (**A**) SunTag reporter schematic. In addition to the tandem SunTag repeats and the auxin-inducible degron, this reporter encodes nano luciferase and BFP, which are not used in the present experiments. The 3' UTR also encodes tandem repeats of the MS2 stem loop, which can be used to label the mRNA red. However, since we detect mRNA by FISH, we do not use the MS2 stem loops in the present experiments. (**B**) Example cells imaged by FISH-IF. Cells were either

*Figure 4 continued on next page*

*Figure 4 continued*

untreated (top row), treated with 91 μM puromycin for 5 min (second row), pre-treated with 208 μM emetine for 15 min followed by 91 μM puromycin for 5 min (third row), or pre-treated with 37 μM anisomycin for 5 min followed by 91 μM puromycin for 5 min (last row). Yellow arrows: examples of translating mRNAs; White arrows: example of single fully synthesized SunTag polypeptide (released from the ribosome); Blue arrows: examples of untranslating mRNAs. Scale bar in top left image: 10 microns. Scale bar in top right image: two microns. (C) Fraction of mRNAs co-localized with SunTag signal. Each dot represents one cell. Cells are only included in the analysis if they have more than five and fewer than 36 mRNAs. 20–27 cells and 313–513 mRNAs per condition were analyzed. Black lines indicate mean with standard error of the mean. P values were calculated by two-sample t-test. Experiment performed once.

The online version of this article includes the following source data for figure 4:

**Source data 1.** Source data for *Figure 4C*.

cells (~20 microns) (*Borle, 1969*), in which the method was demonstrated (*David et al., 2012*). Thus, limiting puromycin treatment time to a few minutes does not ensure that nascent proteins remain confined to the subcellular region in which they are synthesized.

## Discussion

In this work, we have demonstrated that the puromycin method for visualizing localized translation does not faithfully detect nascent polypeptides at the site of their synthesis. Puromycylated polypeptides are released from the ribosome and can diffuse far away from the site of synthesis, even following short treatment times. Additionally, treatment with emetine does not prevent release of puromycylated peptides from the ribosome. This is in agreement with previous polysome gradient analysis (*Colombo et al., 1965*; *Grollman, 1968*) and real-time SunTag imaging (*Wang et al., 2016*) that has shown that neither emetine nor cycloheximide ultimately prevent release of these peptides. It is therefore likely that the specific subcellular localizations detected by this method are in many

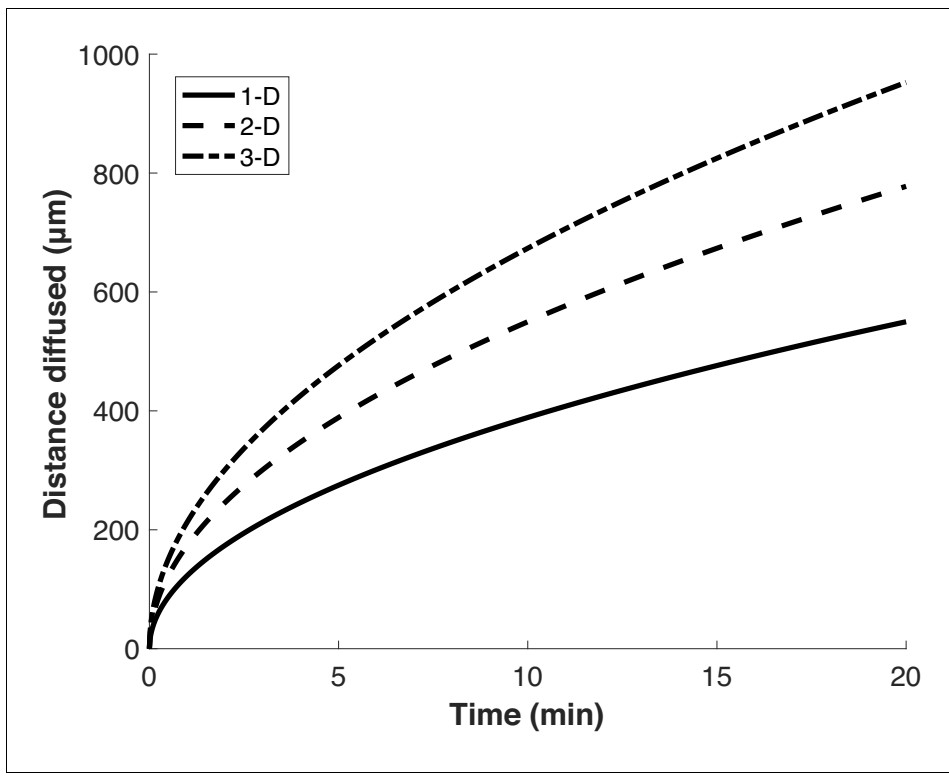

**Figure 5.** Proteins diffuse over long distances in the cell during common puromycin labeling times. Calculation of expected displacement by diffusion as a function of time, using the equation $<x^2> = 2nDt$ where n is the dimensionality, D is the diffusion coefficient (126 μm$^2$/s *Di Rienzo et al., 2014*) and *t* is time. The calculation is shown for 1, 2 and 3 dimensions.

cases the end result of trafficking or diffusion of partially-synthesized puromycylated proteins. For example, the bright nucleolar labeling detected in *David et al., 2012* likely does not reflect nucleolar translation, but the trafficking of puromycylated N-terminal fragments of highly abundant ribosomal proteins (which have N-terminal nucleolar localization signals) to the nucleolus (*Kubota et al., 1999*; *Schmidt et al., 1995*). Thus, conclusions reached using this method should be treated cautiously, even in neurons, where cellular projections protrude relatively far from the cell body. We note that other reporter-based methods that rely on rapid, single turnover chemistry, such as the 'flash' activity of some luciferases, may allow for more accurate localization of the sites of protein synthesis (*Na et al., 2016*). Additionally, in principle, derivatizing puromycin with a chemical moiety large enough to obstruct its passage through the ribosome exit tunnel would immobilize reacted nascent chains on the ribosome. The viability of this concept is demonstrated by the RiboLace method (*Clamer et al., 2018*), which uses a puromycin-biotin conjugate bound to magnetic beads to capture translating ribosomes from cellular lysates. Of course, these beads are not cell permeable and are unsuitable for in vivo imaging. The methods outlined here will be useful for screening cell-permeable puromycin derivatives for their ability to faithfully localize protein synthesis.

# Materials and methods

## Key resources table

| Reagent type (species) or resource | Designation | Source or reference | Identifiers | Additional information |
|---|---|---|---|---|
| Recombinant DNA reagent | pGEM-luc (plasmid) | Promega | GenBank X65316.2 | Firefly luciferase cassette vector |
| Recombinant DNA reagent | pSL312 (plasmid) | This paper | | Full-length firefly luciferase template; can be obtained from Green Lab |
| Recombinant DNA reagent | P3.35_pGEM_luc_trunc_kozak_RC (plasmid) | This paper | | Truncated firefly luciferase template; can be obtained from Green Lab |
| Peptide, recombinant protein | StuI (restriction enzyme) | NEB | R0187S | Linearization of pSL312 for SP6 transcription |
| Peptide, recombinant protein | HpaI (restriction enzyme) | NEB | R0105S | Linearization of P3.35 for SP6 transcription |
| Sequence-based reagent | Full-length luciferase mRNA | This paper | | SP6 transcribed from pSL312 |
| Sequence-based reagent | Truncated luciferase mRNA | This paper | | SP6 transcribed from P3.35 |
| Commercial assay or kit | mMESSAGE mMACHINE SP6 transcription kit | Invitrogen | AM1340 | |
| Commercial assay or kit | Nuclease-treated rabbit reticulocyte lysate translation reactions | Promega | L4960 | |
| Chemical compound, drug | Luciferin | PerkinElmer | 122799 | |

*Continued on next page*

*Continued*

| Reagent type (species) or resource | Designation | Source or reference | Identifiers | Additional information |
|---|---|---|---|---|
| Peptide, recombinant protein | Superase-In RNase Inhibitor | Invitrogen | AM2696 | |
| Chemical compound, drug | 5'-guanylyl imidodiphosphate (GDPNP) | Jena Bioscience | NU-401–50 | |
| Chemical compound, drug | Emetine | Cayman Chemical | 21048 | |
| Chemical compound, drug | Puromycin | Sigma Aldrich | P7255 | |
| Chemical compound, drug | Anisomycin | Sigma | A9789 | |
| Genetic reagent (*C. elegans*) | N2 | Caenorhabditis Genetics Center (CGC) | | |
| Commercial assay or kit | Click-iT Plus OPP Alexa Fluor 488 Protein Synthesis Assay kit | Invitrogen | C10456 | |
| Cell line (human) | U-2OS cells containing Flp-In locus | Andrew Holland lab (Johns Hopkins University) | | |
| Chemical compound, drug | amino-11–12 ddUTP | Lumiprobe | A5040 | |
| Peptide, recombinant protein | deoxynucleotidyl transferase | Thermo Fisher | EP0162 | |
| Chemical compound, drug | doxycycline hyclate | Millipore Sigma | D9891 | |
| Chemical compound, drug | Cy3-NHS ester | Lumiprobe | 41020 | |
| Chemical compound, drug | 3-indole acetic acid | Sigma Aldrich | I2886 | |
| Chemical compound, drug | paraformaldehyde | Electron Microscopy Sciences | 50-980-492 | |
| Peptide, recombinant protein | rat tail collagen I | Gibco | A1048301 | |
| Peptide, recombinant protein | BSA | VWR | VWRV0332-25G | |
| Chemical compound, drug | SSC buffer | Corning | 46–020 CM | |

*Continued on next page*

Continued

| Reagent type (species) or resource | Designation | Source or reference | Identifiers | Additional information |
|---|---|---|---|---|
| Chemical compound, drug | formamide | Sigma Aldrich | F9037-100ML | |
| Sequence-based reagent | *E. coli* tRNA | Sigma Aldrich | 10109541001 | |
| Chemical compound, drug | dextran sulfate | Sigma Aldrich | D8906-100G | |
| Chemical compound, drug | ribonucleoside vanadyl complex | NEB | S1402S | |
| Antibody | Chicken polyclonal anti-GFP antibody | Aves Labs | RRID:AB_2307313 | 1:1000 dilution |
| Antibody | Goat anti-chicken polyclonal IgY secondary antibody | Thermo Fisher | RRID:AB_2534096 | 1:1000 dilution |
| Chemical compound, drug | ProLong Diamond antifade reagent | Invitrogen | P36962 | |
| Recombinant DNA reagent | pubc-OSTIR1-IRES-scFv-sfGFP-NLS (plasmid) | Reference 32 | | |
| Software, algorithm | FISH-quant | Reference 45 | | |
| Software, algorithm | Custom MATLAB scripts for processing FISH-quant output | This paper | | Scripts for quantifying number of translating ribosomes per mRNA from FISH-IF data; available as *Source code 1* in supplementary files |
| Sequence-based reagent | Oligonucleotides used to generate FISH probes | Reference 32 | | See *Supplementary file 1* |

## Plasmid construction

A Kozak consensus sequence (GCCACC) was inserted immediately upstream of the start codon of luciferase reporter plasmid pGEM-luc (GenBank X65316.2) to generate pSL312. This was used as the template for full-length firefly luciferase mRNA transcription and was linearized with a StuI restriction digest. For reasons unrelated to the current work, a disabled 2A peptide sequence was fused downstream of the luciferase sequence to generate P3.28_pGEM_ luc_2A_AGP|_kozak_RC, which was further modified by inserting an HpaI restriction site 2 nt 3′ of the final codon of luciferase (TTGtt|aac, where TTG is the final luciferase sense codon) through site-directed mutagenesis to make P3.35_pGEM_luc_trunc_kozak_RC. This plasmid was linearized with an HpaI restriction digest such that transcription of this template would terminate at TTGtt and would exclude the 2A peptide sequence. For SunTag experiments, the plasmid pcDNA_CMV_ST was used to generate a stable cell line using the Flp-In method. pcDNA_CMV_ST contains an open reading frame coding for 23x SunTag repeats, Nano Luciferase, BFP and an auxin-inducible degron, expressed from a pcDNA5 vector.

## OPP-click

*C. elegans* was cultured according to standard methods at 20 °C. N2 adult germlines were dissected into egg buffer with 1 mM levamisole. Germlines were incubated with 45 µM emetine, or 37 µM anisomycin (*Bastide et al., 2018*), or egg buffer alone for 15 min. OPP was added at a concentration of 20 µM while maintaining concentrations of emetine and anisomycin for the 5 min incubation. Germlines were rinsed once with PBS and fixed in 4% paraformaldehyde. Click reaction was carried out with Click-iT Plus OPP Alexa Fluor 488 Protein Synthesis Assay kit (Thermo Fisher C10456) according to the manufacturer's directions.

## *C. elegans* imaging

Images were taken with a Zeiss Axio Observer equipped with a CSU-W1 SoRA spinning disk scan head (Yokogawa) and Slidebook v6.0 software (Intelligent Imaging Innovations). Germline images are 10 µm z-stacks starting at the bottom of the distal germline with 0.27 µm step size using a 63X objective. Average intensity projections were quantified in ImageJ. An ROI was drawn around the mitotic zone of each germline and fluorescence in the 488 nm channel was measured. Fluorescence intensity of each germline was normalized to the average intensity of the germlines treated with OPP alone.

## Luciferase-based real-time translation monitoring assay

Luciferase plasmids were linearized with a blunt-end restriction enzyme just upstream (truncated) or downstream (full-length) of the stop codon, followed by transcription with the mMESSAGE mMACHINE SP6 transcription kit (Invitrogen AM1340). Synthesized mRNA was quantified using a Nanodrop 1000. Nuclease-treated rabbit reticulocyte lysate translation reactions (Promega L4960) were set up in a 384-well plate (Thermo Scientific 164610) on ice. Luciferin (PerkinElmer 122799) was added to each reaction well to a concentration of 0.5 mM followed by 12 units of Superase-In RNase Inhibitor (Invitrogen AM2696). SP6-transcribed truncated or full-length firefly luciferase mRNA was added to a concentration of 40 µg/mL using a multichannel pipette and the plate was immediately inserted into a luminometer microplate reader (Biotek Synergy H1MD) regulated at 30°C. Luminescence readings were taken every few seconds, depending on the number of reaction wells. 5'-guanylyl imidodiphosphate (GDPNP; Jena Bioscience NU-401–50) was added to the wells 16 min after the start of the reaction for 5 min at a concentration of 100 µM followed by a 5-min pretreatment of either 208 µM emetine (Cayman Chemical 21048) or 9.4 µM anisomycin (Sigma A9789). Puromycin (Sigma Aldrich P7255) was added to wells at a concentration of 91 µM. In experiments where GDPNP was not used, the first translation inhibitors were added to the reaction wells at 21 min following the start of the reaction. Reagents were added to the wells by first ejecting the microplate from the luminometer and pipetting the reagents in using a multichannel pipette. The microplate was then promptly inserted again.

## Stable cell line

U-2OS cells stably expressing the SunTag reporter were generated using the Flp-In system with the pcDNA_CMV_ST plasmid, as described in *Goldman et al., 2020*. The cell line was a kind gift from Dr. Andrew Holland (Johns Hopkins University). While the cell line's identity has not recently been authenticated via STR profiling, it has frequently been tested for mycoplasma contamination and is mycoplasma free. smFISH Probe Labeling smFISH probes targeting the SunTag region of the mRNA reporter transcript were synthesized as described (*Goldman et al., 2020*; *Gaspar et al., 2017*). 20-mer oligonucleotides (*Supplementary file 1*) were ordered from IDT in an arrayed format, pooled, and labeled on the 3'-end with amino-11–12 ddUTP (Lumiprobe A5040) using deoxynucleotidyl transferase (TdT, Thermo Fisher EP0162). After size exclusion purification on a Spin-X centrifuge column (Corning 8161) with Bio Gel P-4 Beads (Bio Rad 1504124), the oligonucleotide was labeled with Cy3-NHS ester (Lumiprobe 41020). Following the labeling reaction, the probes were again purified over a Spin-X column to remove excessive dyes.

smFISH-IF smFISH-IF was performed similarly as described (*Goldman et al., 2020*; *Latallo et al., 2019*). smFISH-IF was performed on U-2OS cells stably expressing the SunTag mRNA reporter and scFV-sfGFP. 18 mm #1 coverslips (Fisher 12-545-100) were etched in 3M sodium hydroxide (Millipore Sigma 221465) prior to cell plating. The coverslips were then washed 3x with PBS (Corning 21–031-

CV) and then coated for 30 min at 37°C with 0.25 mg/mL rat tail collagen I (Gibco A1048301) diluted in 20 mM sodium acetate (Sigma-Aldrich S2889). After another 2x PBS wash, 18,000 cells were plated per well and grown for 24 hr in DMEM supplemented with 10% FBS. 24 hr following plating, the media was supplemented with 1 μg/mL doxycycline hyclate (Millipore Sigma, #D9891) and 500 μM 3-indole acetic acid (IAA) (Sigma-Aldrich I2886).

Approximately, 24 hr following induction, cells were treated with either 91 μM puromycin in the medium for 5 min, 208 μM emetine in the medium for 15 min followed by 91 μM puromycin in the medium for 5 min, or 37 μM anisomycin in the medium for 5 min followed by 91 μM puromycin in the medium for 5 min. Control cells were left untreated. Following treatment, samples were prepared for smFISH-IF. All solutions were prepared in nuclease free water (Quality Biological 351-029-131CS). Cells were washed 3x with 1x PBS (Corning 46–013 CM) + 5 mM magnesium chloride (Sigma-Aldrich M2670-500G) (PBSM). Cell were then fixed for 10 min at room temperature in PBSM + 4% paraformaldehyde (Electron Microscopy Sciences 50-980-492). Following fixation, samples were washed for $3 \times 5$ min in PBSM and permeabilized for 10 min in PBSM + 5 mg/mL BSA (VWR VWRV0332-25G) + 0.1% Triton-X100 (Sigma-Aldrich T8787-100mL). After $3 \times 5$ min washes in PBSM, cells were incubated for 30 min at room temperature in 2xSSC (Corning 46–020 CM), 10% formamide (Sigma-Aldrich F9037-100ML), and 5 mg/mL BSA (VWR VWRV0332-25G). Following pre-hybridization incubation, samples were incubated for 3 hr at 37°C in 2xSSC (Corning 46–020 CM), 10% formamide (Sigma-Aldrich F9037-100ML), 1 mg/mL competitor *E. coli* tRNA (Sigma-Aldrich 10109541001), 10% w/v dextran sulfate (Sigma-Aldrich D8906-100G), 2 mM ribonucleoside vanadyl complex (NEB S1402S), 100 units/mL SUPERase In (Thermo Fisher AM2694), 60 nM SunTag_v4-Cy3 smFISH probes, and 1:1000 chicken anti-GFP (Aves Labs GFP-1010). The coverslips were then washed 4x with 2xSSC (Corning 46–020 CM) + 10% formamide (Sigma-Aldrich F9037-100ML). The samples were then incubated with $2 \times 20$ min with a goat anti-chicken IgY secondary antibody labeled with Alexa Fluor 488 (Thermo Fisher A-11039). After 3x washes in 2xSCC, cells were mounted on pre-cleaned frosted glass cover slides (Fisher 12-552-3) with ProLong Diamond antifade reagent with DAPI (Invitrogen P36962). After curing for 24 hr, the samples were imaged on a custom Nikon Ti-2 wide-field microscope equipped with a $60 \times 1.4$ NA oil immersion objective lens (Nikon), a Spectra X LED light engine (Lumencor), and an Orca 4.0 v2 scMOS camera (Hamamatsu). The microscope was under automated control by Nikon Elements software. x-y pixel size: 107.5 nm. z-step: 300 nm.

## smFISH Analysis

Fixed cell image analysis was performed as described (*Goldman et al., 2020*) with existing or custom MATLAB software. Spot detection of the mRNA and proteins channels were performed independently in FISH-Quant (*Mueller et al., 2013*). In the protein channel, all released single peptides in the cytoplasm were detected and thresholded based on their Gaussian fitting parameters (intensity and width) and inspected to ensure accuracy. All released single peptides were then averaged into an idealized point spread function to calculate the integrated intensity of a single SunTag array. In the mRNA channel, only cytoplasmic RNAs were included for analysis. After determining all cytoplasmic mRNA positions, FISH-Quant's transcription site quantification algorithm was employed to quantify the integrated intensity of the associated translation site. Briefly, a $11 \times 11$ bounding box was drawn at the position of each mRNA and Gaussian fitting was performed centered on the brightest pixel within this box. The integrated intensity of the translation site was then normalized against the intensity of the idealized single peptide to calculate the number of nascent chains associated with a given mRNA. The translation sites were filtered based on shape, intensity, and distance from the mRNA. Failure to converge on an accurate fit given these parameters resulted in the associated translation site intensity to have an intensity value of 0. Translation sites with an integrated intensity of less than one idealized single peptide were determined to be unassociated with SunTag signal. Only cells with greater than five and fewer than 35 mRNAs were considered.

## Acknowledgements

This work was funded by the National Institutes of Health [2R37GM059425-14 to RG; 5R37HD037047-20 to GS; 5K99GM135450-02 to BZ] and the Howard Hughes Medical Institute (HHMI) (RG and GS). BZ was an HHMI fellow of the Damon Runyon Cancer Research Foundation

[DRG-2250–16] for a portion of this study. DHG is a Damon Runyon Fellow supported by the Damon Runyon Cancer Research Foundation (DRG-2280–16). NML and MC are supported by NIH Training Grant T32 GM007445. MC is supported by National Science Foundation Graduate Research Fellowship DGE-1746891. Some *C. elegans* strains were provided by the CGC, which is funded by NIH Office of Research Infrastructure Programs (P40 OD010440)

## Additional information

### Competing interests

Rachel Green: Reviewing editor, *eLife*. The other authors declare that no competing interests exist.

### Funding

| Funder | Grant reference number | Author |
|---|---|---|
| National Institutes of Health | 2R37GM059425-14 | Rachel Green |
| National Institutes of Health | 5R37HD037047-20 | Geraldine Seydoux |
| National Institutes of Health | 5K99GM135450-02 | Boris Zinshteyn |
| Howard Hughes Medical Institute | | Geraldine Seydoux Rachel Green |
| Damon Runyon Cancer Research Foundation | DRG-2280-16 | Daniel H Goldman |
| Damon Runyon Cancer Research Foundation | DRG-2250-16 | Boris Zinshteyn |
| National Institutes of Health | T32 GM007445 | Madeline Cassani Nathan M Livingston |
| National Science Foundation | DGE-1746891 | Madeline Cassani |

The funders had no role in study design, data collection and interpretation, or the decision to submit the work for publication.

### Author contributions

Syed Usman Enam, Investigation, Writing - original draft; Boris Zinshteyn, Conceptualization, Supervision, Methodology, Writing - original draft, Writing - review and editing; Daniel H Goldman, Conceptualization, Investigation, Writing - original draft, Writing - review and editing; Madeline Cassani, Conceptualization, Investigation, Writing - original draft; Nathan M Livingston, Formal analysis, Methodology; Geraldine Seydoux, Conceptualization, Writing - review and editing; Rachel Green, Conceptualization, Supervision, Writing - original draft, Writing - review and editing

### Author ORCIDs

Syed Usman Enam ⓘD https://orcid.org/0000-0001-8976-0660
Boris Zinshteyn ⓘD https://orcid.org/0000-0003-0103-3501
Nathan M Livingston ⓘD http://orcid.org/0000-0003-4670-708X
Geraldine Seydoux ⓘD http://orcid.org/0000-0001-8257-0493
Rachel Green ⓘD https://orcid.org/0000-0001-9337-2003

### Decision letter and Author response

Decision letter https://doi.org/10.7554/eLife.60303.sa1
Author response https://doi.org/10.7554/eLife.60303.sa2

## Additional files

### Supplementary files

• Source code 1. Script for analysis of FISH-IF data.

- Supplementary file 1. Oligonucleotide sequences used to generate smFISH probes.
- Transparent reporting form

### Data availability

Raw data for all plots have been deposited in accompanying excel files.

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
