## [Decision Letter]

**Acceptance summary:**

The work convincingly shows that puromycin releases nascent polypeptides even in the presence of elongation inhibitors. This observation questions interpretations using detection of puromycin to denote sites of translation.

**Decision letter after peer review:**

Thank you for submitting your article "Puromycin reactivity does not accurately localize translation at the subcellular level" for consideration by *eLife*. Your article has been reviewed by three peer reviewers, one of whom is a member of our Board of Reviewing Editors, and the evaluation has been overseen by David Ron as the Senior Editor. The following individuals involved in review of your submission have agreed to reveal their identity: Marvin Tanenbaum (Reviewer #2); Maria Ugalde (Reviewer #3).

The reviewers have discussed the reviews with one another and the Reviewing Editor has drafted this decision to help you prepare a revised submission.

Summary:

The work presented here convincingly shows that the puromycylated signal comes from released, truncated polypeptides, and not the nascent chains associated with the ribosome. In a straightforward experiment, using the SunTag system, the puromycylated nascent chains were not present on ribosomes, but only after the peptides were released from the ribosome did the fluorescence of the tagged puromycylated peptides become evident indicating that puromycin released them immediately upon incorporation even in the presence of elongation inhibitors.

Essential revisions:

The reviewers have different suggestions-one feels that the literature needs to be more fully referenced. One other suggests consideration of the statistical basis for the observations in Figure 2 (has it been repeated?) and that the use of cycloheximide would have made the manuscript more robust. Acceptance of the manuscript does not require additional experimentation however, but include your response to these comments in a revised manuscript.

Reviewer #1:

The authors of this, and the co-submitted manuscript investigated the assumption that puromycin is a spatial indicator of translating ribosomes. This assumption is based on the puromycylation of nascent chains remaining associated with the ribosome on the mRNA, in effect freezing in its position there in the presence of elongation inhibitors. When detected by antibodies to the puromycin derivative, the signal supposedly indicated the sites of translation.

The work presented here convincingly shows that the puromycylated signal comes from released, truncated polypeptides, and not the nascent chains associated with the ribosome. In a straightforward experiment, using the suntag system, the puromycylated nascent chains were not present on ribosomes, but only after the peptides were released from the ribosome did the fluorescence of the tagged puromycylated peptides become evident indicating that puromycin released them immediately upon incorporation even in the presence of elongation inhibitors.

The work is of significance particularly since a number of high impact publications have relied on the puromycylation signal indicating the location of ongoing translation. Conclusions that derive from that assumption have to be re-evaluated, since some of them are clearly wrong (e.g. translation in the nucleus) or misleading (work on neurons). I feel it is important to publish this work before more erroneous conclusions contaminate the literature.

Reviewer #2:

This study by Enam and colleagues re-exams a frequently used assay based on puromycin incorporation to determine the sub-cellular site of mRNA translation. They convincingly show that puromycin treatment results in rapid release of nascent polypeptides, and that this is not inhibited by the translation elongation inhibitor emetine. I agree with the authors that it is important to point out this flaw in the puromycin incorporation assay, and their study does an excellent job in doing this. The main issue here is that there is already data in the literature that shows that nascent chains rapidly (tens of seconds) dissociate from the ribosome and move away from the site of translation, even in the presence of a translation elongation inhibitor (CHX). One example that comes to mind is the paper from the Zhuang lab (PMID: 27153499), where they show that nascent chains are fully released in 1-2 min upon addition of puro and CHX (note that they do actually observe a small delay in nascent chain release due to CHX). That being said, the Zhuang lab paper does not make a point out of this result and how it affects interpretation of OPP assays, and I believe that is still useful.

Reviewer #3:

Green's group questions the accuracy of the ribopuromycylation method (RPM) to report on the subcellular localization of translation. In RPM, translation elongation inhibitors (emetine and cyclohexamide) preclude the release of puromycylated peptides from the ribosome. Using three approaches, they demonstrate that emetine does not prevent the release of nascent peptides, which diffuse from the ribosome after puromycin treatment: 1) in vitro translation assay of luciferase reporters, 2) Localization of O-propargyl-puromycin (OPP) elongating peptides in the nucleus of *C. elegans* cells and, 3) Lack of co-localization of the SunTag translation reporter with its encoded mRNA in mammalian cells. Authors warn on the use of puromycylation assays as the sole approach to localize translation in cells.

Findings from this work will certainly help the scientific community to interpret the data from puromycylation experiments and to choose the best experimental approaches to assess the spatial regulation of translation. The authors state that experiments have been performed once or twice. Hence, the existing experimental evidence could benefit from additional data and analysis to support the main conclusion (even though a similar manuscript submitted to this journal by the group of Peter A. Seems shows similar findings).

1) Only emetine was used to inhibit translation elongation. Adding cyclohexamide to some or all the experiments will validate their conclusions.

2) Experiment in Figure 2 has been done only once. The same experiment should be repeated two more times in worms.

3) In Figure 3, the media of two experiments is plotted, but the standard deviation is not indicated. The same experiment should be repeated one more time. The experiment in Figure 3—figure supplement 1B has done only once, but it is presented as a major conclusion in the Results section of the paper.

4) Figure 4. The number of cells considered for the quantifications is low for a fixed experiment (two experiments of ten cells each). The experiment should be repeated one more time, and authors are encouraged to increase the number of cells they quantify.

5) Figure 5 is a simple calculation of diffusion to support the idea that puromycylated peptides can be detected far from their site of translation. Two suggestions to support conclusions from this part of the paper: 1) Improve the mathematical modeling to fit translation localization results obtained from puromycylated experiments in neurons and cell lines, and 2) Support the conclusions from the modeling with experimental data. The latest can be done using their SunTag reporter system in U2OS cells or neurons to track single peptides.

---

## [Author Response]

Essential revisions:The reviewers have different suggestions-one feels that the literature needs to be more fully referenced. One other suggests consideration of the statistical basis for the observations in Figure 2 (has it been repeated?) and that the use of cycloheximide would have made the manuscript more robust. Acceptance of the manuscript does not require additional experimentation however, but include your response to these comments in a revised manuscript.

In response to reviewer comments, we have clarified a number of statements and references in our Introduction and Discussion to provide broader context and clarify methods. We have updated Figure 3 to clarify the range of data obtained from replicate experiments and added data from an additional replicate to Figure 2.

In addition, the example untreated cell in Figure 4B was replaced with a more representative image, since the previous image had more mRNAs (~40) than was allowed in our analysis (35), thus making it unrepresentative of the cells that were analyzed. We have also increased the size of the zoom panels in this figure and normalized the red channel to the same values to increase clarity.

Reviewer #2:This study by Enam and colleagues re-exams a frequently used assay based on puromycin incorporation to determine the sub-cellular site of mRNA translation. They convincingly show that puromycin treatment results in rapid release of nascent polypeptides, and that this is not inhibited by the translation elongation inhibitor emetine. I agree with the authors that it is important to point out this flaw in the puromycin incorporation assay, and their study does an excellent job in doing this. The main issue here is that there is already data in the literature that shows that nascent chains rapidly (tens of seconds) dissociate from the ribosome and move away from the site of translation, even in the presence of a translation elongation inhibitor (CHX). One example that comes to mind is the paper from the Zhuang lab (PMID: 27153499), where they show that nascent chains are fully released in 1-2 min upon addition of puro and CHX (note that they do actually observe a small delay in nascent chain release due to CHX). That being said, the Zhuang lab paper does not make a point out of this result and how it affects interpretation of OPP assays, and I believe that is still useful.

We thank the reviewer for putting our study in the broader context of other work in the field. We agree that evidence for release of nascent chains after puromycin addition in the presence of cycloheximide or emetine inhibitors has existed for decades. To our knowledge, this was first shown by Colombo in 1965 and Grollman in 1968; in particular, the Grollman paper showed that there was at most very modest stabilization of nascent peptide on elongating ribosomes in the presence of emetine, while Colombo demonstrated that cycloheximide only transiently stabilized nascent peptides on the ribosome. We also appreciate that the group of single molecule imaging papers (looking at translating ribosomes) emerging several years ago used puromycin release of polypeptides as an experiment to identify elongating ribosomes in real time (Yan et al., 2016; Wang et al., 2016; Morisaki et al., 2016; Wu et al., 2016; Pichon et al., 2016). The paper from the Zhuang lab referenced by the reviewer (Wang et al., 2016) indeed shows a short delay in release of nascent peptide by puromycin in the presence of CHX, but it is not clear if this (as well as the effect seen by Colombo in 1965) is due to stabilization of puromycylated peptides on the ribosome, or slowed reactivity of puromycin with the unusual ribosome state induced by CHX (Budkovitch et al., Mol Cell 2011, PMID 22017870). Regardless of the mechanism, this slight stabilization would still ultimately result in complete dissociation of nascent peptide on the time scales used in puromycylation imaging experiments. Despite this literature, there remains widespread usage of “puromycylation” to localize elongating ribosomes, prompting us to make a specific point of this matter. In particular, given that the work by Yewdell and colleagues made a strong and specific point about the role of emetine in stabilizing puromycylated peptides on ribosomes, we felt it important to specifically address this point. We have added several sentences in the Discussion to more fully acknowledge the literature in this area.

Reviewer #3:[…] 1) Only emetine was used to inhibit translation elongation. Adding cyclohexamide to some or all the experiments will validate their conclusions.

We chose to perform our experiments with emetine due to its much slower off-rate and the more common use of this inhibitor in studies utilizing RPM. Indeed, in the initial development of this approach, it was argued that emetine was more useful for stabilizing nascent peptides on ribosomes for this reason (David et al., 2012), though some recent studies utilizing RPM did not include emetine. We expect that CHX would produce similar results as seen in previous studies using metabolic labeling (Colombo 1965) and SunTag live imaging (Wang et al., 2016).

2) Experiment in Figure 2 has been done only once. The same experiment should be repeated two more times in worms.

We have repeated this experiment again to validate our initial observations.

3) In Figure 3, the media of two experiments is plotted, but the standard deviation is not indicated. The same experiment should be repeated one more time. The experiment in Figure 3—figure supplement 1B has done only once, but it is presented as a major conclusion in the Results section of the paper.

We have updated the graphs in Figure 3 and Figure 3—figure supplement 1B with shading to indicate the full range of values obtained in the replicates. We also clarify that the experiment in Figure 3—figure supplement 1B was performed twice as noted in the figure legend. While we appreciate the fact that repeating an experiment in an identical manner makes the result of a single experiment more robust, the RRL experiments reported in Figure 3B, C and Figure 3—figure supplement 1B are fundamentally similar and were performed independently, and all support the same conclusion that emetine does not prevent release of puromycylated peptides from the ribosome. In light of current limitations on experimental time, we argue that sufficient robustness derives from these similar experiments.

4) Figure 4. The number of cells considered for the quantifications is low for a fixed experiment (two experiments of ten cells each). The experiment should be repeated one more time, and authors are encouraged to increase the number of cells they quantify.5) Figure 5 is a simple calculation of diffusion to support the idea that puromycylated peptides can be detected far from their site of translation. Two suggestions to support conclusions from this part of the paper: 1) Improve the mathematical modeling to fit translation localization results obtained from puromycylated experiments in neurons and cell lines, and 2) Support the conclusions from the modeling with experimental data. The latest can be done using their SunTag reporter system in U2OS cells or neurons to track single peptides.

The data we obtained from our several fixed imaging experiments are highly consistent with one another and we feel provide sufficient statistical significance to conclude that puromycylated peptides are not retained on the elongating ribosomes, with or without emetine.

To address the second point, our modeling parameters are in fact derived from measured diffusion constants for GFP, so we don’t think there is much value in making these measurements for our SunTag peptides. This modeling exercise was simply meant to provide some sense of the time scales for diffusion that would be relevant to experiments routinely reported in the literature.